# Static Stability of Composite Annular Plates with Auxetic Properties

**DOI:** 10.3390/ma15103579

**Published:** 2022-05-17

**Authors:** Dorota Pawlus

**Affiliations:** Faculty of Mechanical Engineering and Computer Science, University of Bielsko-Biala, 43-308 Bielsko-Biala, Poland; doro@ath.bielsko.pl

**Keywords:** auxeticity, buckling, composite annular plate, finite element method

## Abstract

This paper presents an evaluation of the static stability of complex, composite annular plates with layers having auxetic properties. The main objective of the numerical investigations is the development of a plate model, which uses an approximate solution based on orthogonalization and finite difference methods. The three-layered plate is composed of auxetic facings and a soft, foam core. The material properties of the facings are characterized by Poisson’s ratio, the values of which are variously positive and negative. The results obtained for an auxetic plate were compared on the basis of the results for a plate with traditional facings and a plate model built of finite elements. Additionally, in order to verify the calculation results, an analysis of the homogeneous plate was performed. Two plate models built of finite differences and finite elements were compared. The wide image of buckling responses of the examined plates was created on the basis of the calculation results of both compressed and radially stretched plates. An increase in the values of the critical static loads with increasing absolute value of Poisson’s ratio of auxetic facings is one of the observations.

## 1. Introduction

The phenomenon of auxeticity exhibited by materials with a negative Poisson’s ratio is practically important and scientifically inspiring during the evaluation of the mechanics of composite structures. Analysis of the mechanical behaviours of auxetic materials and a special complex group of materials—multi-materials in which one of the constituents is an auxetic material—has been undertaken in numerous works.

The approach to estimating the material parameters and evaluating the stress state is also a subject of consideration. Refs. [1,2,3,4,5] can serve as an example.

Composite materials such as periodic laminates built of auxetic and isotropic elastic constituents are the subject of the consideration by Ramirez et al. [1]. By analysing various material combinations, the influence of the laminate structure on the total mechanical properties of the composite was shown. An analysis of the laminated periodic composite material with the use of the homogenization technique was presented by Donescu et al. [2]. The value of Young’s modulus, which was found to be equal to 1550 MPa for the auxetic materials on the basis of the simulation process, was adopted for the exemplary analyses in this paper. Analytical, finite element and experimental approaches to the modelling of auxetic cell foam liners and curved auxetic thermoplastic plates were presented by Mohanraj et al. [3]. Experimental and computational investigations evaluating the normal and shear behaviours of the mechanically attractive group of materials known as auxetic metamaterials, which also exhibit compressional resistance, were presented by Henyš et al. [4]. Due to the existence of a group of materials with negative values of Poisson’s ratio, it was found to be necessary to estimate the values of the elastic moduli for the classical linear isotropic theory of elasticity. The proposed solution based on the condition of the positivity of the internal energy of the deformation of the material was presented by Rushchitsky [5].

Employing auxetic materials in classic sandwich structures can create new possibilities for the usage of such structures, the properties of which are specific. Their significant rigidity and strength in connection with their light weight are the main known qualities of sandwich structures. The incorporation of the auxetic layer can change the mechanical response of the composite. The circular and annular plates of a sandwich structure with orthotropic composite facings and an auxetic core were examined by Alipour and Shariyat [6]. Two methods were proposed: one based on the principle of the minimum total potential energy and the other using the ABAQUS programme. Numerous results have been reported presenting the distributions of lateral deflections and in-plane normal and transverse shear stresses. This highlights the effect of the auxeticity of the plate core. It also shows the influence of the rigidity of the plate structure, the type of support system, and the geometric dimensions.

A three-layered sandwich plate with an auxetic core and isotropic, homogeneous facings was presented in [7]. The influence of the geometric, boundary and load parameters on the static bending of the investigated plate was analysed. A bending analysis of FGM shell and plate structures with auxetic properties was presented by Shariyat and Alipour [8]. The problem was solved analytically and numerically, with numerous results being presented for different combinations of geometrical, material, loading and support parameters.

The final conclusions included some observations that were similar to those presented in this paper, like those related to the influence of increasing the value of Poisson’s ratio to a value equal to *ν* = −0.9. The optimal Poisson’s ratio for minimizing the bending stress of the auxetic circular plate was determined in Lim’s work [9]. Strongly dependent elements such as the deformed shape, the load distribution, and edge supports were taken into consideration. Studies for the analysis of maximum deflection were carried out investigating an axisymmetrical example of a plate with different values of Poisson’s ratio assuming three cases: constant value of Young’s modulus, constant value of Kirchhoff’s modulus, and constant value of the product of Young’s and Kirchhoff’s moduli. The buckling and vibration analysis of the auxetic circular plate was presented by Lim [10]. The critical parameters were calculated for an axisymmetric plate made of isotropic material with values of Poisson’s ratio within the range of −1 to 0.5.

In classic sandwich structures, where the facings are loaded with normal stresses but the core is loaded with shear stresses, the influence of Poisson’s ratio occurs when describing the material parameters of the facings. This kind of structure is adopted for the investigations presented in this paper, the main aim of which is the evaluation of the effect of negative values of Poisson’s ratio. Numerical analyses are performed for a three-layered annular plate, as the research object, subjected to loads acting in the plane of the plate facings. Critical parameters like loads and modes, which characterize the static state of plate buckling, are subjected to detailed analysis. An example of work in which eigenvalue buckling and post-buckling analyses are performed is that of Faghfouri et al. [11]. The thin circular disks are subjected to two edge loads. The stability behaviour of disks made from both traditional, linear elastic and auxetic materials under compression and tension loading was evaluated. The values of Poisson’s ratio were varied over a wide range. Great differences were observed between disks subjected to compression and tension in the numerical results.

In this paper, numerical investigations were carried out in two tracks for two plate models. The first model was obtained by solving the problem analytically and numerically, but the second one was built using finite elements. The presented numerical calculation results for the exemplary plates provide an image of the static plate responses. The evaluation of the buckling behaviour of the plate composed of three layers made of two different materials, one traditional and the second with conventional parameters expressed by a positive value of Poisson’s ratio or with unconventional parameters determined by a negative value of Poisson’s ratio, significantly complements our understanding of the annular plates under examination. The problem of the stability of annular plates built using traditional elastic or viscoelastic layers has been addressed in numerous works, and has been rather widely examined. Works by Chen and Wang et al. [12,13] can serve as examples. Various areas of application for composite annular plates have been underlined, for example, in the aerospace industry, in mechanical and nuclear engineering, in civil engineering, or in miniature mechanical systems, and the continuing search for new mechanical possibilities arising from the use of composite structures acts as the natural motivator for undertaking such investigations. The following problems constitute the research questions addressed in this paper: the evaluation of the basic, static critical state of the annular plate with outer layers with auxetic properties; the analysis of the responses of two similar plate models; and the behaviours of homogeneous and heterogeneous layered plates subjected to the radial compression and tension. The numerical calculation results, diagrams, analyses, observations and remarks provided make it possible to formulate relevant answers and demonstrate the properties of auxetic annular plate structures with respect to the buckling issue.

## 2. Problem and Object of the Analysis

Buckling analysis is the main problem that has been addressed. A composite, three-layered annular plate loaded mechanically on the inner or outer edge was the object of analysis. The plate structure was transversely symmetrical composed of thin facings and a thicker foam core. The plate’s outer layers were linearly elastic and traditional, or were made of a special material, the properties of which corresponded to the parameters of auxetic material. The plate transversal structure, with an auxetic–foam–auxetic arrangement, created a structure with some new qualities. Understanding the stability behaviours and the reactions of the plate to mechanical loading enables the evaluation of the possibilities of producing such mixed structures with different material constants. Various cases of plate structures built using auxetic and conventional elastic facings with values of Poisson’s ratio *ν* within the range −0.9 ≤ *ν* ≤ 0.3 were examined in order to supplement the good existing knowledge of sandwich composite plates. The analyses were carried out for a fixed, constant value of Young’s modulus. Depending on the relation between the engineering constants pertaining for the isotropic material, the value of Kirchhoff’s modulus changes. The critical states of composite plates with and without auxetic properties were compared by analysing two buckling models: the classical model, with radial compression; and the opposite, with radial stretching, referred to as “buckling at stretching”.

The scheme of the examined plate is shown in Figure 1. Both plate edges are slidably clamped, meaning that displacement in the radial direction is possible. Thin facings are loaded in the middle plane. The forces are uniformly distributed on the inner or outer plate perimeter. The objective of the numerical investigations and analysis of results was to obtain the values of static and critical loads and their corresponding modes.

## 3. Strategy for Solving the Problem

The proposed analytical and numerical method is a means of solving the problem of the static stability of the plate. A detailed description of classical sandwich plates with conventional facings was presented by Pawlus [14,15,16]. This solution was based on the eigenvalue problem, where the numerically calculated minimal values of loads correspond to the critical static loads. Numerical calculations were carried out using a programme developed by the author.

### 3.1. Main Elements of the Solution

The fundamental elements of the solution are as follows:-The derivation of the equilibrium equations for each plate layer;-The establishment of geometrical relations to express the core deformation with the use of the classical theory of the broken line hypothesis;-The usage of linear physical relations presented by Hooke’s law;-The determination of relations between sectional forces, moments and stresses in the plate facings, transverse force in the core, and the resultant radial and circumferential forces by the stresses function;-The description of the boundary conditions for both slidable clamped edges of the plate;-The establishment of the basic form of the differential equation describing the plate deflections;-The solution procedure based on the dimensionless quantities and parameters, application of the orthogonalization method to eliminate the circumferential angular variable, and usage of the finite difference method (FDM) to replace the derivatives with respect to *ρ* by the finite central differences in the discrete points;-The formulation of the main equation assuming that the stress function is a solution to the disk state:
(1)$$MP\cdot U+MCD\cdot D+MCG\cdot G=p^{*}MC\cdot U$$
where

*p**—dimensionless stress, $p^{*}=\frac{p}{E}$, *p*—radially compressing or stretching stress, *E*—Young’s modulus of the facing material;

***MP***, ***MCD***, ***MCG***, ***MC***—matrices of components constructed from the geometric and material parameters of the plate and quantity *b* (*b*—length of the interval in the finite difference method);

***U***, ***D***, ***G***—vectors expressed by the plate deflections and coefficients *δ* or *γ* (*δ*, *γ*—differences in radial or circumferential displacements at the middle surface of the facings);
-the numerical solution of the eigenvalue problem for calculating the critical, static stress *p_cr_* is as follows:
(2)$$det\left(\left(MP+MCDG\right)-p^{*}MC\right)=0$$
where ***MCDG***—matrix of components constructed from the elements of the matrices ***MCD*** and ***MCG*** and components of reverse matrices constructed from the elements of the matrices expressed by the additional equilibrium equations in the radial and circumferential facing directions (see Equations (9) and (10)).

### 3.2. Meaning of the Sign of Poisson’s Ratio in the Equations of the Three-Layered Annular Plate

Some of the solution elements described in Section 3.1 will be presented in detail to show the effect of the Poisson’s ratio *ν* on the final results of the calculations for annular composite plates with auxetic and conventional facings.

The linear physical relations of Hooke’s law are fundamental, as they determine the influence of the value and the sign of Poisson’s ratio *ν* on the stress–strain state of the facing material of the plate. Physical relations are presented in radial and circumferential plate directions:(3)$$\sigma_{r}=\frac{E}{1-\nu^{2}}(\varepsilon_{r}+\nu \varepsilon_{\theta }),\;\sigma_{\theta }=\frac{E}{1-\nu^{2}}(\varepsilon_{\theta }+\nu \varepsilon_{r}).$$

The relations derived between sectional forces, moments and stresses in the plate facings are presented by the forces *N_r_*, *N_θ_* and the moments *M_r_*, *M_θ_* by means of equations in the radial and circumferential directions, respectively. Poisson’s ratio *ν* occurs as an element in plate rigidity *D* or as a number in Equations (4)–(7):(4)$$N_{r}=\frac{Eh^{\prime }}{1-\nu^{2}}\left(u_{,r}+\frac{1}{2}{\left(w_{,r}\right)}^{2}+\frac{\nu }{r}u+\frac{\nu }{r}v_{,\theta }+\frac{\nu }{2r^{2}}{\left(w_{,\theta }\right)}^{2}\right)$$
(5)$$N_{{}_{\theta }}=\frac{Eh^{\prime }}{1-\nu^{2}}\left(\frac{1}{r}u+\frac{1}{r}v_{,\theta }+\frac{1}{2r^{2}}{\left(w_{,\theta }\right)}^{2}+\frac{1}{2}\nu {\left(w_{,r}\right)}^{2}+\nu \;u_{,r}\right)$$
(6)$$M_{r}=-D\left(w_{{}_{,rr}}+\frac{\nu }{r}w_{{}_{,r}}+\frac{\nu }{r^{2}}w_{{}_{,\theta \theta }}\right)$$
(7)$$M_{\theta }=-D\left(\frac{1}{r^{2}}w_{{}_{,\theta \theta }}+\frac{1}{r}w_{{}_{,r}}+\nu \;w_{{}_{,rr}}\right)$$
where

$D=\frac{{Eh^{\prime }}^{3}}{12\left(1-\nu^{2}\right)}$—plate rigidity of facing material; *w*—plate deflection; *u*, *v*—displacements of the points of the middle plane of facings in the radial and circumferential directions, respectively; *h′*—facing thickness.

The main differential equation that defines the plate deflections includes the value of *ν* in the parameters *K*_1_, *K*_2_:(8)$$\begin{array}{l} K_{1}w_{,rrrr}+\frac{{2K}_{1}}{r}w_{,rrr}-\frac{K_{1}}{r^{2}}w_{,rr}+\frac{K_{1}}{r^{3}}w_{,r}+\frac{K_{1}}{r^{4}}w_{,\theta \theta \theta \theta }+\frac{2\left(K_{1}+K_{2}\right)}{r^{4}}w_{,\theta \theta }+\\ +\frac{{2K}_{2}}{r^{2}}w_{,rr\theta \theta }-\frac{{2K}_{2}}{r^{3}}w_{,r\theta \theta }-G_{2}\frac{H^{\prime }}{h_{2}}\frac{1}{r}\left(\gamma_{,\theta }+\delta +r\delta_{,r}+H^{\prime }\frac{1}{r}w_{,\theta \theta }+H^{\prime }w_{,r}+H^{\prime }rw_{,rr}\right)\\ =\frac{2h^{\prime }}{r}(\frac{2}{r^{2}}\Phi_{,\theta }w_{,r\theta }\;-\;\frac{2}{r}\Phi_{,r\theta }w_{,r\theta }+\frac{2}{r^{2}}w_{,\theta }\Phi_{,r\theta }-\frac{2}{r^{3}}\Phi_{,\theta }w_{,\theta }+w_{,r}\Phi_{,rr}+\Phi_{,r}w_{,rr}+\frac{1}{r}\Phi_{,\theta \theta }w_{,rr}+\frac{1}{r}\Phi_{,rr}w_{,\theta \theta }) \end{array}$$
where

*K*_1_ = 2*D*, *K*_2_ = 4*D_rθ_* + *νK*_1_, $D_{r\theta }=\frac{{{Gh}^{\prime }}^{3}}{12}$; *G*, *G*_2_—Kirchhoff’s modulus of facing and core materials, respectively; *h*_2_—core thickness; *H′* = *h′* + *h*_2_, *δ = u*_3_ − *u*_1_, *γ* = *v*_3_ − *v*_1_; *Φ*—stress function.

Additionally, *ν* is also present in the differential equilibrium equations of forces in the plane of the facings in the radial and circumferential directions:(9)$$\frac{2r}{h_{2}}G_{2}H^{\prime }w_{,r}=\frac{Eh^{\prime }}{1-\nu^{2}}\left(r\delta_{,rr}+\delta_{,r}-\frac{1}{r}\delta +\nu \gamma_{,r\theta }-\frac{1}{r}\gamma_{,\theta }\right)+Gh^{\prime }\frac{1}{r}\left(\delta_{,\theta \theta }+r\gamma_{,r\theta }-\gamma_{,\theta }\right)-\frac{2r}{h_{2}}G_{2}\delta$$
(10)$$\frac{2}{h_{2}}G_{2}H^{\prime }w_{,\theta }=\frac{Eh^{\prime }}{1-\nu^{2}}\frac{1}{r}\left(\delta_{,\theta }+r\nu \delta_{,r\theta }+\gamma_{,\theta \theta }\right)-\frac{2r}{h_{2}}G_{2}\gamma +Gh^{\prime }\frac{1}{r}\left(\delta_{,\theta }+r\delta_{,r\theta }+r^{2}\gamma_{,rr}+r\gamma_{,r}-\gamma \right)$$

Changing the sign of *ν* influences the value of parameter *K_2_* (8) and the quantity $\nu \frac{Eh^{\prime }}{1-\nu^{2}}$ (9), (10). Additionally, differences in the value of Poisson’s ratio *ν* affect the values of Kirchhoff’s modulus *G*. However, in the case of the axisymmetrical plate mode *m* = 0, when derivatives with respect to angle *θ* do not exist ($\frac{\partial }{\partial \theta }=0$), the sign of *ν* will not affect the final results. Additionally, the difference in the displacement in the circumferential direction *γ* of the plate will not influence the final results obtained by means of a solution process in which the following form of the shape function of quantity *γ* is accepted:(11)γ(r,θ)=γ(r)sin(mθ)

In summary, it can be observed that for the fixed, constant value of Young’s modulus, the negative sign of the Poisson’s ratio *ν* affects the results of the asymmetrical *m* ≠ 0 modes of plates with auxetic facings.

## 4. Exemplary Results

In this section, the calculation results will be presented for the exemplary plates with determined geometrical and material parameters in order to show the effect of auxeticity when only present in the material of the facings.

The dimensions of the three-layered annular plate is defined as follows: inner radius *r_i_* = 0.2 m, outer radius *r_o_* = 0.5 m, facing thickness *h′* = 1 or 2 mm, core thickness *h*_2_ = 5 mm. The facings are made of a conventional linear elastic material with a Poisson’s ratio *ν* = 0.3, a material with a Poisson’s ratio *ν* = 0, or an auxetic material with a negative value of Poisson’s ratio *ν* = −0.3, −0.6, −0.9. The Young’s modulus of the materials of the facings is fixed, and is a constant *E* = 1550 MPa [2]. Due to the relation for the isotropic material *G* = *E*/2(1 + *ν*), the value of Kirchhoff’s modulus *G* for the facings changes from *G* = 596.15 MPa for *ν* = 0.3 to *G* = 7750 MPa for *ν* = −0.9. Polyurethane foam with Kirchhoff’s modulus *G*_2_ = 5 MPa is the core material, and is treated as an isotropic material. Additionally, the examined homogeneous annular plate is assumed to have the same geometric, material and support parameters as the single outer layer of the annular plate of the sandwich.

### 4.1. Convergence Evaluation

Table 1, Table 2 and Table 3 present the convergence evaluation of the critical static loads *p_cr_* corresponding to the plate modes *m* for the selected number of discrete points *N* using the finite difference method. The number *N* creates a set of discrete values for all continuous functions, which are replaced by the corresponding operators of the differences in the values of the functions for the selected points *N*.

These results are for auxetic plates radially compressed on the outer or inner edge and radially stretched on the inner edge with Poisson’s ratio *ν* =−0.6, −0.9, and −0.3, respectively. The analyses were carried out for the chosen numbers *N* equal to 11, 14, 17, 21, and 26. Observation of the presented numbers enables the formulation of the following remarks:-With increasing plate mode, the differences between the corresponding loads *p_cr_* become smaller;-The minimal values of the critical static loads *p_cr_* are obtained for the asymmetrical plate mode (the bold numbers) with several circumferential waves (*m* = 5, 7) and the axisymmetric (*m* = 0) mode of plates radially compressed on the inner edge;-Appropriate accuracy, determined on the basis of a technical error of up to 5%, is ensured by using a number *N* equal to 14, which was thus used in subsequent numerical calculations.

### 4.2. Composite Annular Plate Radially Compressed

Figure 2 and Figure 3 show the critical static load *p_cr_* distribution as a function of the value of Poisson’s ratio *ν*. Figure 2 presents the results for plates loaded on the outer perimeter, while Figure 3 presents the results obtained for plates compressed on the inner edge. Results are presented for the following values of Poisson’s ratio: *ν* = 0.3, 0.0, −0.3, −0.6, and −0.9. The values of *p_cr_* and the modes *m* with which they correspond change similarly for the each of examined plates, the facing materials of which are described on the basis of their accepted positive or negative values of Poisson’s ratio. Figure 2 shows the results for plates with two considered values of facing thickness *h′* = 1 mm and *h′* = 2 mm. The minimum value of critical static load is found for plate mode *m* = 5. Plates with thicker auxetic facings, the absolute valuesare of Poisson’s ratio of which is also higher, *ν* = −0.9, loses stability with a number of circumferential waves *m* = 6. Then, the value of *p_cr_* reaches its minimum. With increasing absolute value of Poisson’s ratio in auxetic facings, the critical static load *p_cr_* increases. The values of *p_cr_* for plates with thicker facings are higher for the axisymmetical *m* = 0 mode and the asymmetrical *m* ≠ 0 mode with several circumferential waves. For higher numbers of waves *m* (*m* = 9, *m* = 10), small changes in the values of *p_cr_* for plate cases with *h′* = 1 mm and *h′* = 2 mm can be observed (see Figure 2c). The differences between the values of *p_cr_* for plates with *h′* = 2 mm are smaller. The dependence of *p_cr_* distribution on the number *m* is flatter than what can be observed for the plate with *h′* = 1 mm. Plates compressed on the inner edge (see Figure 3) lose their static stability in the axisymmetrical form of buckling *m* = 0. Mode *m* = 0 corresponds to the minimum value of critical static load *p_cr_*. The effect of the auxetic facings does not change the previously observed regularity of the axisymmetrical form *m* = 0 of plate buckling [14,15,16]. The presented results show that there are no significant differences between the results obtained for the plates with auxetic (*ν* = −0.3) and traditional (*ν* = 0.3) facing materials (see, Figure 2d and Figure 3). When the composite plate structure is built using a thicker foam elastic core, the significance of the auxetic facings is eliminated.

In conclusion, it can be observed that the use of facings with auxetic parameters used to build plates with classic sandwich structures does not change the buckling reactions compared to plates composed of layers made of conventional materials.

### 4.3. Composite Annular Plate Radially Stretched

The problem of the loss of stability can also be observed for annular plates subjected to forces that stretch the plate in a radial direction. This is an issue that has rather seldom been addressed. The examined composite annular plate is loaded with forces regularly distributed on the inner plate edge. The forces are directed to the middle of the plate (see Figure 1b). Figure 4 presents the distribution of the values of critical static loads *p_cr_* as a function of the mode number of the plates with various values of Poisson’s ratio.

The minimum values of *p_cr_* can be observed for the asymmetrical form of buckling *m* = 7. The buckling mode changes when the absolute value of negative Poisson’s ratio is increased to *ν* = −0.9. Then, the number *m* is equal to *m* = 14, and the differences between the values of *p_cr_* for the higher plate modes (*m* > 10) become smaller. Higher mode numbers *m* of auxetic plates with a high value of absolute negative Poisson’s ratio *ν* = −0.9 can be clearly observed under buckling caused by radial stretching, in contrast to with radial compression. Increased values of *p_cr_* can also be observed with incredasing absolute values of Poisson’s ratio *ν*.

In conclusion, it can be stated that the phenomenon of “buckling at stretching” also confirms the regular character of the plate responses. The stability of the reactions are similar among the examined composite plates with facings with smaller absolute positive or negative values of Poisson’s ratio *ν*. There are fewer differences in the values of the critical loads *p_cr_*.

## 5. Comparative Analysis

The comparison of the selected results of the numerical calculations of the FDM and FEM plate models enables the verification of the calculations, as well as further observations of plate behaviours. A comparison was performed between the examined heterogeneous three-layered annular plate and the homogeneous plate.

The finite element method was applied as a second means of conducting numerical investigations. Calculations were performed in the ABAQUS 2022 system (product of Dassault Systemes Simulia Corp., Johnston, RI, USA) at the Academic Computer Centre CYFRONET-CRACOW (KBN/SGI_ORIGIN_2000/PLodzka/030/1999). Using the Buckle option of the ABAQUS programme, the static eigenvalue solution was determined [17]. The three-layered FEM plate model was built of shell and solid elements to produce the facings and the core meshes, respectively. The homogeneous FEM plate model was built using shell elements. The facing mesh and the mesh of the homogeneous plate were composed of 3D nine-node shell elements. The core mesh of the heterogeneous plate was built using 3D 27-node solid elements. The three-layered and homogeneous plate models took the form of a full annulus. Surface contact interaction was applied to tie the surfaces of the facing meshes to the surfaces of the core mesh.

### 5.1. Homogeneous Annular FDM and FEM Plate Models

A solution to the problem of deflections in homogeneous annular plates can be obtained by eliminating Equations (9) and (10) and modifying the main Equation (8). The single-plate outer layer was described by dividing both sides of Equation (8) by the number 2, and the single-plate outer layer with thickness *h’* was accepted as the layer of the homogeneous plate with thickness *h* (*h* = *h′*) under the assumption that core thickness did not exist, *h*_2_ = 0; thus, the new form of Equation (8) presents the main equation of the solution process of the homogeneous plate deflections, as follows:(12)$$\begin{array}{l} \frac{1}{2}K_{1}w_{,rrrr}+\frac{K_{1}}{r}w_{,rrr}-\frac{K_{1}}{2r^{2}}w_{,rr}+\frac{K_{1}}{2r^{3}}w_{,r}+\frac{K_{1}}{2r^{4}}w_{,\theta \theta \theta \theta }+\frac{\left(K_{1}+K_{2}\right)}{r^{4}}w_{,\theta \theta }+\frac{K_{2}}{r^{2}}w_{,rr\theta \theta }-\frac{K_{2}}{r^{3}}w_{,r\theta \theta }=\\ \frac{h^{\prime }}{r}(\frac{2}{r^{2}}\Phi_{,\theta }w_{,r\theta }\;-\;\frac{2}{r}\Phi_{,r\theta }w_{,r\theta }+\frac{2}{r^{2}}w_{,\theta }\Phi_{,r\theta }-\frac{2}{r^{3}}\Phi_{,\theta }w_{,\theta }+w_{,r}\Phi_{,rr}+\Phi_{,r}w_{,rr}+\frac{1}{r}\Phi_{,\theta \theta }w_{,rr}+\frac{1}{r}\Phi_{,rr}w_{,\theta \theta }) \end{array}$$

Figure 5, Figure 6 and Figure 7 show the distribution of the values of the critical static loads *p_cr_* of the homogeneous FDM and FEM annular plate models with a thickness *h* = 2 mm.

Figure 8 shows exemplary modes of the homogeneous plate with auxetic facings with Poisson’s ratio *ν* = −0.3 when radially stretched on the inner edge.

The results in the cases of the plates radially compressed on the outer and inner edge are presented in Figure 5 and Figure 6, respectively. The results of the analysis of the “buckling at stretching” problem are presented in Figure 7. A good compatibility can be observed between the minimum values of the critical loads *p_cr_* of the FDM and FEM plate models. Greater differences exist for higher plate modes. The involvement of the auxetic material in the homogeneous plate results in relevant differences in the values of loads *p_cr_* between the material with the value of Poisson’s ratio *ν* = −0.9 and the materials with other values.

Summarizing the presented results of auxetic and conventional homogeneous plates, the following can be observed:Lower values of critical loads *p_cr_* are routinely found for the FDM plate model than the FEM model, which has importance for stability analysis;There was significant growth in the values of critical loads *p_cr_* with increasing absolute number of Poisson’s ratio *ν* for the auxetic plate, as also observed in [8];There was a lack of differences in responses between plates with auxetic material with Poisson’s ratio *ν* = −0.3 and conventional material with Poisson’s ratio *ν* = 0.3;Importantly, the plate stability problem was solved, by taking into account asymmetric buckling modes. As it is more simple, the axisymmetric (*m* = 0) solution can be applied only for plates radially compressed on the inner edge.

### 5.2. Three-Layered Annular FDM and FEM Plate Models

A comparison of the calculation results obtained using the FDM and FEM plate models is shown in Figure 9 and Figure 10. The calculation results for the plates radially compressed on the outer edge are presented in Figure 9.

Figure 9a shows the results obtained using the finite element method for plates with auxetic properties. The distribution of values of the critical load *p_cr_* as a function of the mode *m* confirms the calculations performed for plates with traditional facings. A significant increase in the value of *p_cr_* can be observed for plates with auxetic facings with a high value of Poisson’s ratio *ν* = −0.9. The minimum value of critical load *p_cr_* can be found for mode *m* = 5 for the plate with a value of Poisson’s ratio *ν* = −0.9 and for mode *m* = 4 for the other plates. A comparison of the *p_cr_* distribution for the FDM and FEM plate models with two exemplary values of Poisson’s ratio *ν* = −0.3 and *ν* = −0.6 is presented in Figure 9b. The values of *p_cr_* calculated using the finite element method are lower than those obtained using the finite difference method. With increasing values of the number *m*, the difference decreases to values in the range of 1 MPa.

Figure 10 shows the reaction of the FDM and FEM plate models when radially stretched on the inner edge. The calculation results of the critical loads *p_cr_* of the FDM and FEM plate models are presented for two material structures characterized by values of Poisson’s ratio *ν* = 0 and *ν* = −0.3. Similarly to the case of plates radially compressed on the outer edge, the values of the critical loads *p_cr_* calculated for the FEM model are lower than those calculated for the FDM plate model. Additionally, the difference in values becomes smaller with increasing number *m*. 

Figure 11 shows exemplary modes of the three-layered plate with auxetic facings with Poisson’s ratio *ν* = −0.6, radially compressed on the outer edge.

Summarizing the presented results, the following observations can be made:There is an agreement between the characteristics of the responses of the FDM and the FEM plate models with auxetic and traditional properties, in which high values of critical loads *p_cr_* for plates with auxetic facings with a high Poisson’s ratio value *ν* = −0.9 can also be observed (see Figure 2a and Figure 9a);Plates radially compressed on the outer perimeter and stretched on the inner one exhibit asymmetric buckling;The structure of the plate model built with finite elements possesses greater flexibility, the core mesh of which is composed of solid elements and does not eliminate the existing effect of bending.

## 6. Conclusions

The analysis of the classic sandwich structure of annular plates with auxetic properties was the subject of the buckling investigations undertaken in this study. Negative values of Poisson’s ratio, which characterise auxetic materials, influence the structural responses through the character of work on the outer plate layers when subjected to normal stresses. The numerical calculation results for the critical static loads of plates when both compressed and stretched in the radial direction were presented in this paper. Thus, the stability case denoted as “buckling at stretching” was also evaluated. For the material parameters selected for the facings, the behaviours of plates compressed and stretched on the inner edge in the radial direction were compared. The results were shown for plates with facings made of auxetic materials, as well as those made from convential linear elastic materials. The evaluation of the critical state of the plates was considered, while taking into account the effect of the various values of Poisson’s ratio. Additionally, the comparative analysis of the calculation results of two three-layered FDM and FEM annular plate models—homogeneous and heterogeneous—was conducted. A number of detailed remarks regarding the analysed plate with slidably clamped edges were formulated. Among these, the following should be distinguished:For the fixed, constant value of the Young’s modulus, a negative sign of Poisson’s ratio *ν* affects the results of the asymmetrical *m* ≠ 0 modes of plates with auxetic facings;The asymmetric form of buckling applies to plates radially compressed on the outer perimeter and stretched on the inner one;There was an increase in critical static loads *p_cr_* with increasing absolute values of Poisson’s ratio in auxetic facings;The impact of the value of Poisson’s ratio, particularly the examined value *ν* = −0.9, which is close to the number −1, was observed;For the corresponding results obtained for Poisson’s ratio *ν* = −0.6, the increase in the values of static loads for auxetic plates with a limiting value of Poisson’s ratio *ν* = −0.9 reaches more than 60% for plates with the axisymmetric form of buckling, and more than 40% for plates with asymmetric modes;The effect of the auxetic facings does not change the buckling reaction of plates with conventional facings (cf., the results for *ν* = −0.3 and *ν* = 0.3).

It should be noted that with respect to stability, the worst results were observed for the plates with a value of Poisson’s ratio *ν* = 0, where the critical static loads were minimal.

A full analysis of the responses of the examined plates requires a generalised solution that includes asymmetric modes of plate buckling. The proposed approximate analytical and numerical solution, whose fundamentals have been used in calculations related to traditional plates [14,15,16], enables the effective investigation of new plate structures. Complex, multi-parameter tasks, which constitute a problem for the stability of three-layered annular plates, can be successfully analysed by taking into account various plate parameters: geometrical, material and loading. A comparative analysis of the calculation results obtained for two plate models, FDM and FEM, in particular for homogeneous plates (see Section 5.1 of this paper), confirms the efficiency of the analytical and numerical solution. In the case of three-layered plates, this method of modelling the core layer differentiates between the FDM and FEM models. In the analytical and numerical solutions the plate core was loaded only with shear stresses. In the FEM model, it was not possible to exclude the loading with normal stress, on the basis of which it can be stated that the flexibility of the FEM model is better. The characteristics of the responses of both plate models were similar, for both traditional plates and auxetic ones. Differences were observed for auxetic plates in terms of greater absolute values of negative Poisson’s ratio. The presented summary provides answers for the research questions formulated in the Introduction, supported by the calculations and results analyses performed.

The presented approach for evaluating the critical parameters of composite annular plates indicates the sensitivity of the examined structure to the auxetic, inverse mechanical mechanism of the strains, which are in component facings. The analysed composite structure, auxetic–foam–auxetix, was shown to have buckling responses characterized by predictable regularity. However, the numerical analyses, supported by experimental research, should be continued. In further numerical analyses, the dynamic stability problem of annular plates with auxetic properties will be examined.

## Figures and Tables

**Figure 1 materials-15-03579-f001:**
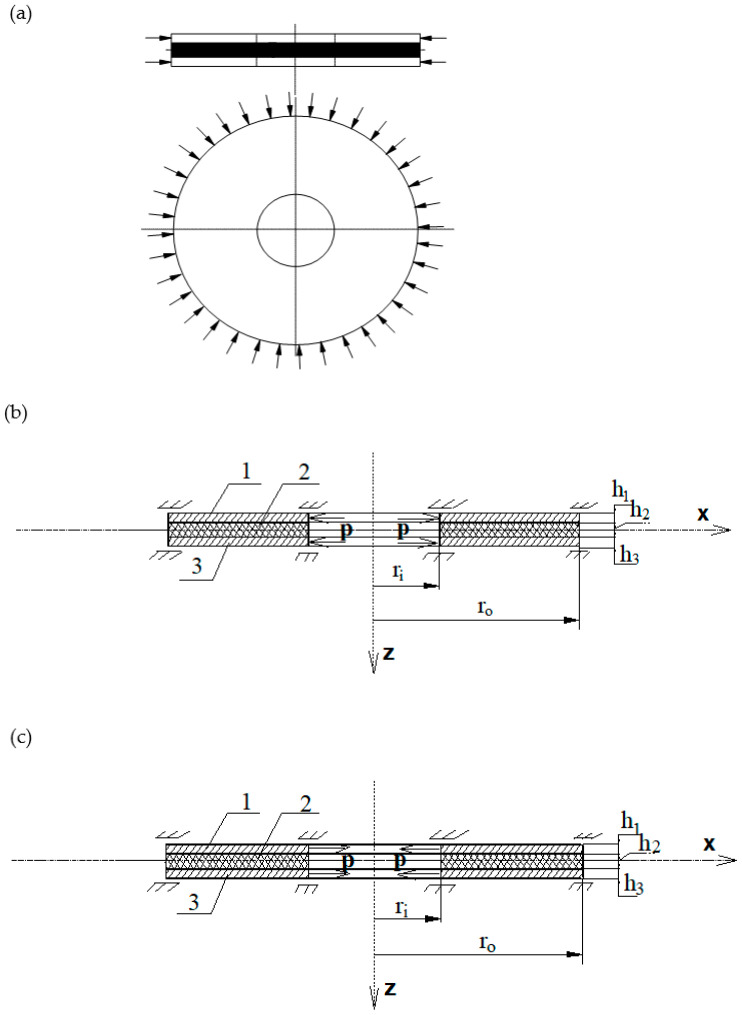
The scheme of the plate radially compressed on the outer edge (**a**). Cross-sections of the plates (1,3—outer layers, 2—core) when radially compressed on the inner edge (**b**), and radially stretched on the inner edge (**c**).

**Figure 2 materials-15-03579-f002:**
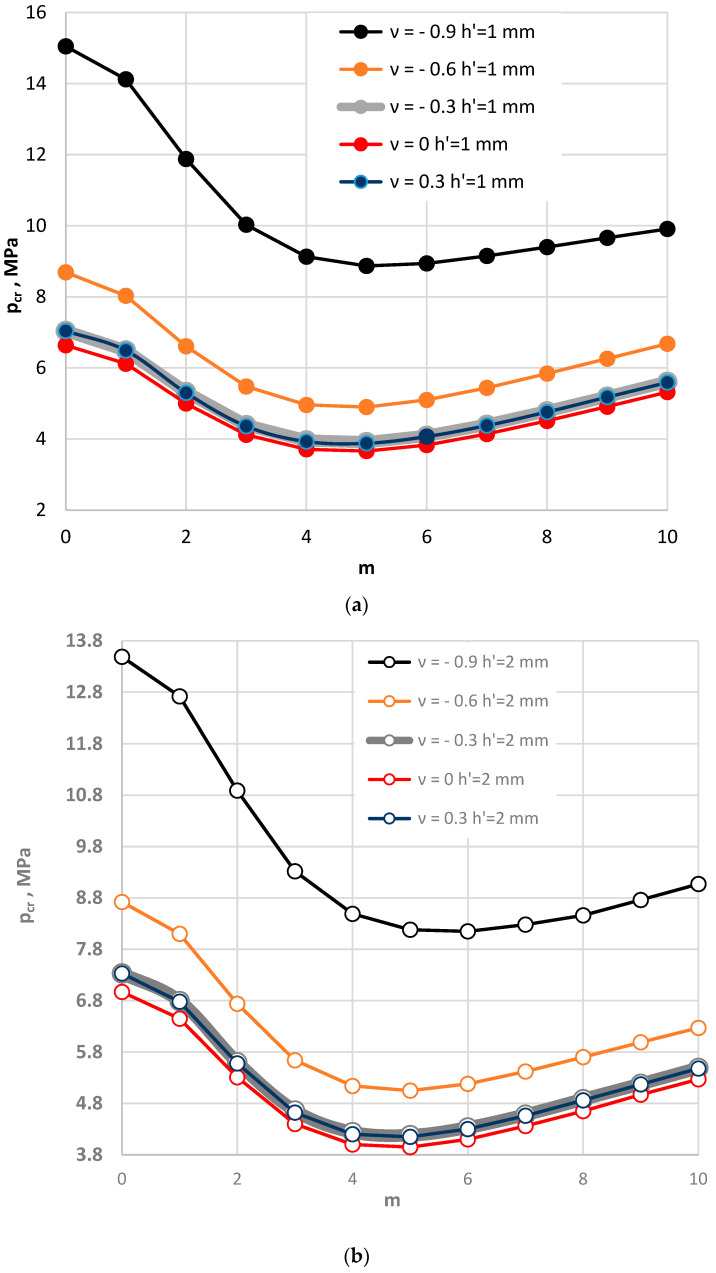
Critical static load *p_cr_* distribution versus mode *m* for plates radially compressed on the outer edge: with facing thickness *h′* = 1 mm (**a**), with facing thickness *h′* = 2 mm (**b**), with two thicknesses of auxetic facings (**c**), with auxetic and conventional facings (**d**).

**Figure 3 materials-15-03579-f003:**
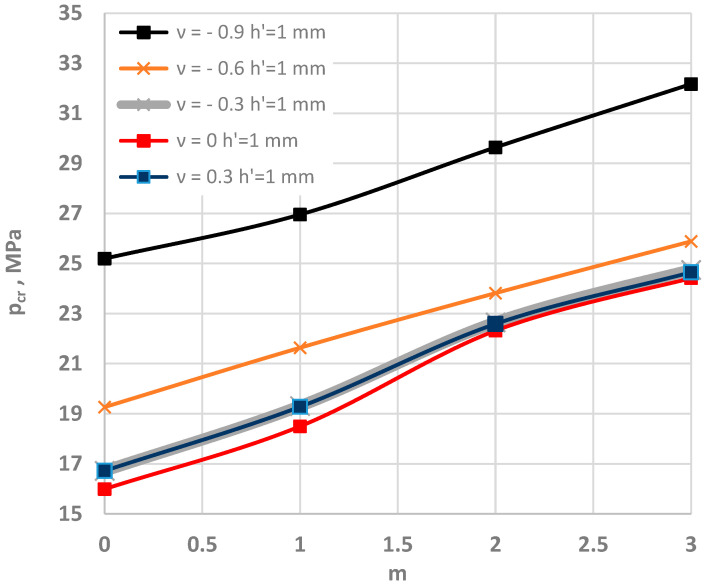
Critical static load *p_cr_* distribution versus mode *m* for plates radially compressed on the inner edge.

**Figure 4 materials-15-03579-f004:**
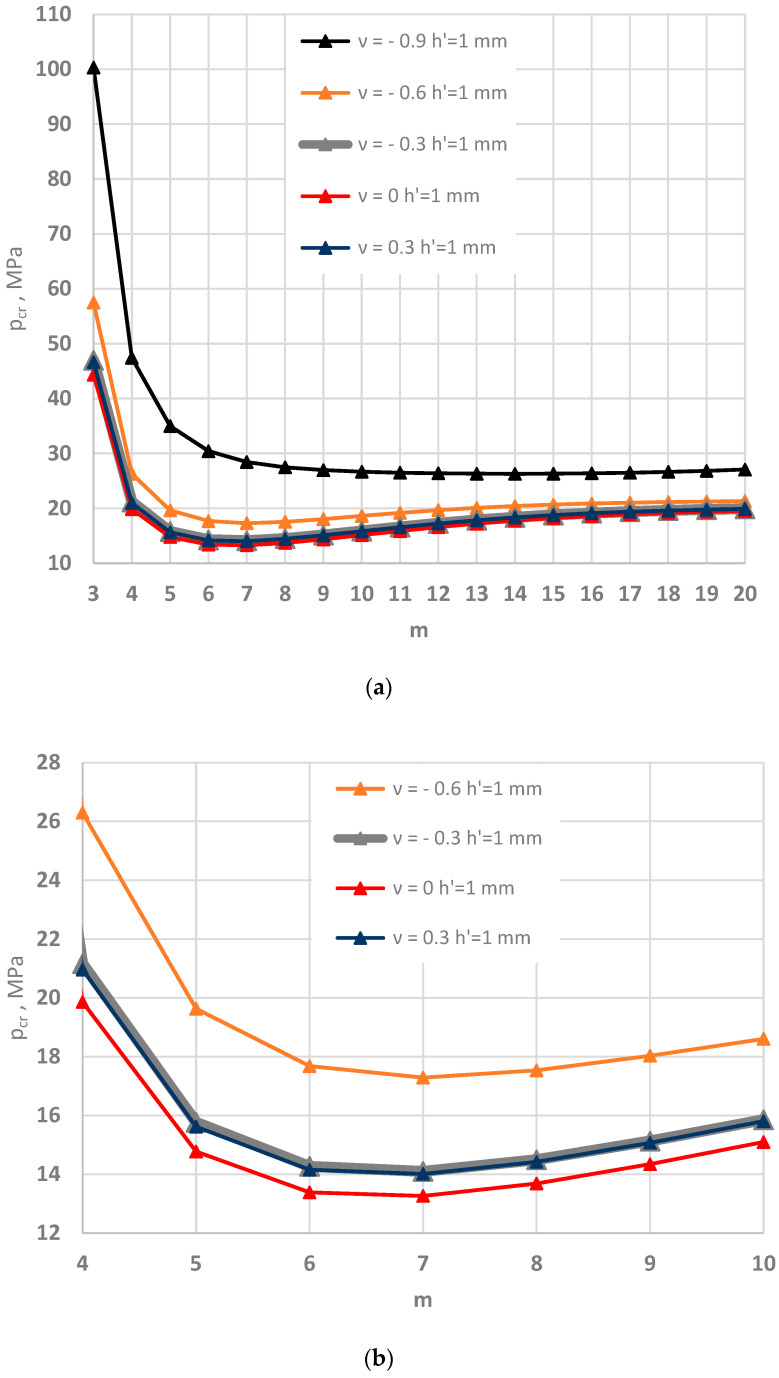
Critical static load *p_cr_* distribution versus mode *m* for plates radially stretched on the inner edge: for values of Poisson’s ratio −0.9 ≤ *ν* ≤ 0.3 (**a**); for values of Poisson’s ratio −0.6 ≤ *ν* ≤ 0.3 (**b**).

**Figure 5 materials-15-03579-f005:**
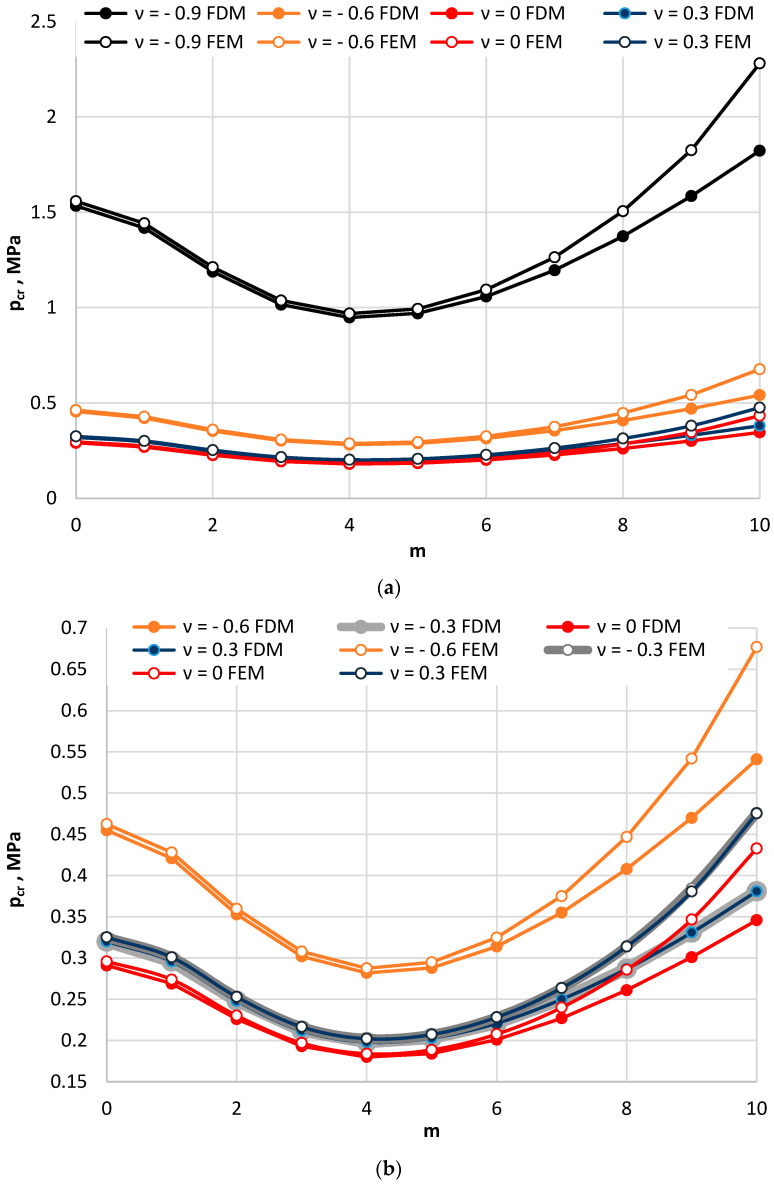
Critical static load *p_cr_* distribution versus mode *m* for the FDM and FEM homogeneous plate models radially compressed on the outer edge: for values of Poisson’s ratio *ν* = −0.9, −0.6 and *ν* = 0, 0.3 (**a**), for values of Poisson’s ratio −0.6 ≤ *ν* ≤ 0.3 (**b**).

**Figure 6 materials-15-03579-f006:**
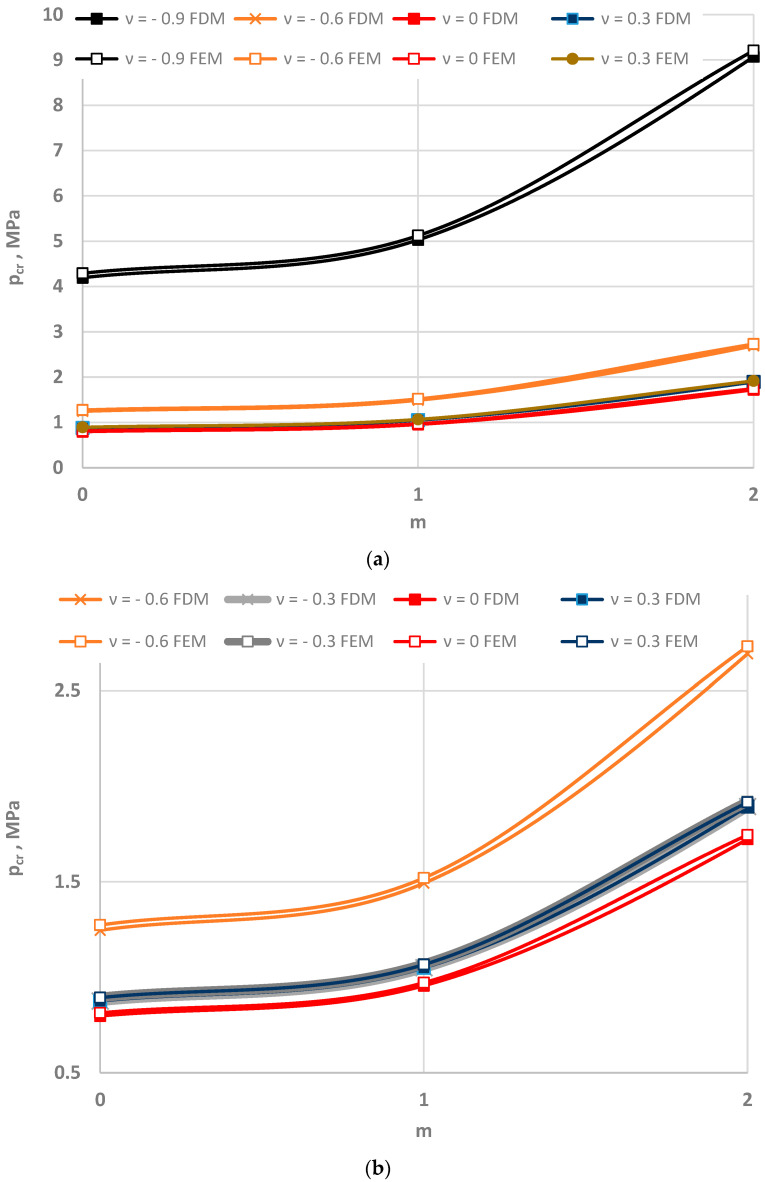
Critical static load *p_cr_* distribution versus mode *m* for the FDM and FEM homogeneous plate models radially compressed on the inner edge: for values of Poisson’s ratio *ν* = −0.9, −0.6 and *ν* = 0, 0.3 (**a**), for values of Poisson’s ratio −0.6 ≤ *ν* ≤ 0.3 (**b**).

**Figure 7 materials-15-03579-f007:**
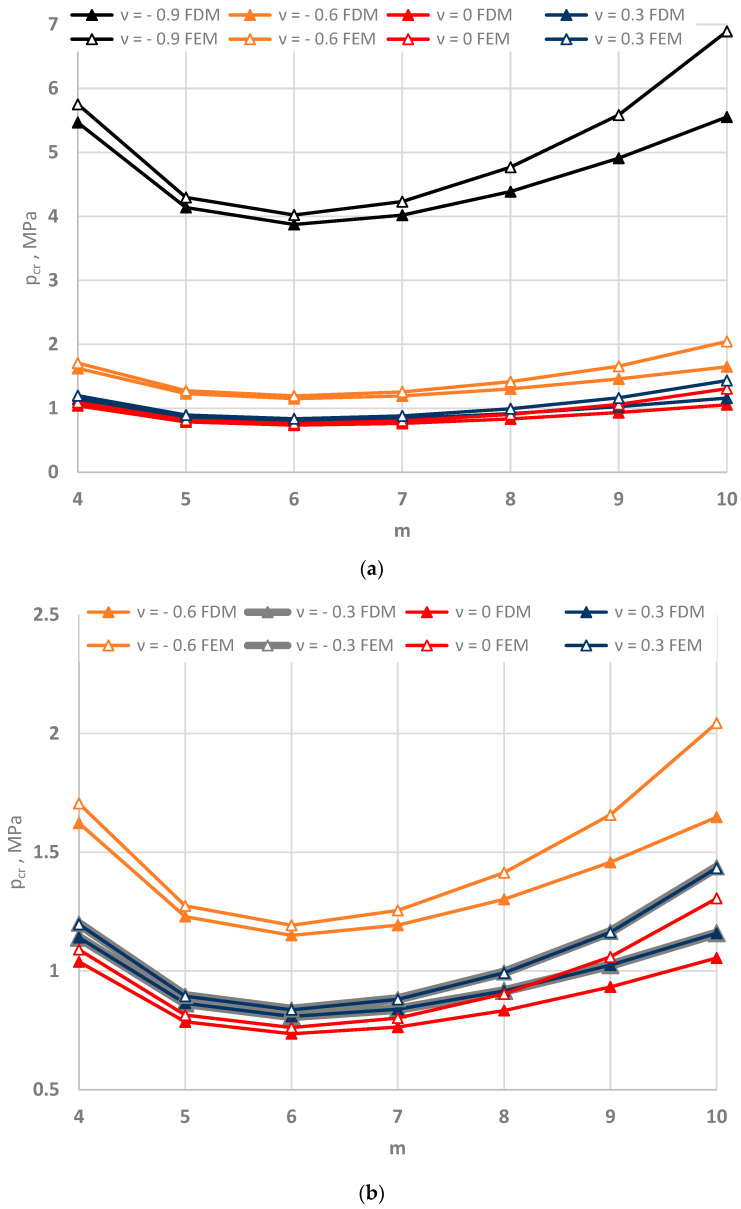
Critical static load *p_cr_* distribution versus mode *m* for the FDM and FEM homogeneous plate models radially stretched on the inner edge: for values of Poisson’s ratio *ν* = −0.9, −0.6 and *ν* = 0, 0.3 (**a**), for values of Poisson’s ratio −0.6 ≤ *ν* ≤ 0.3 (**b**).

**Figure 8 materials-15-03579-f008:**
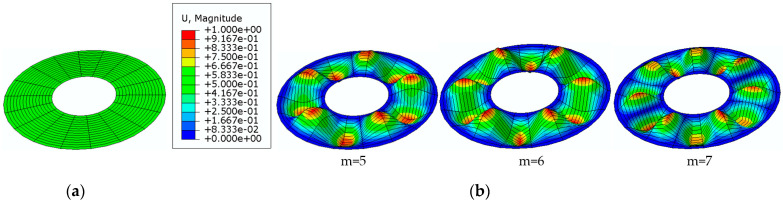
Homogeneous annular FEM plate model: undeformed view (**a**), modes *m* of the plate radially stretched on the inner edge with auxetic facings with Poisson’s ratio *ν* = −0.3 (**b**).

**Figure 9 materials-15-03579-f009:**
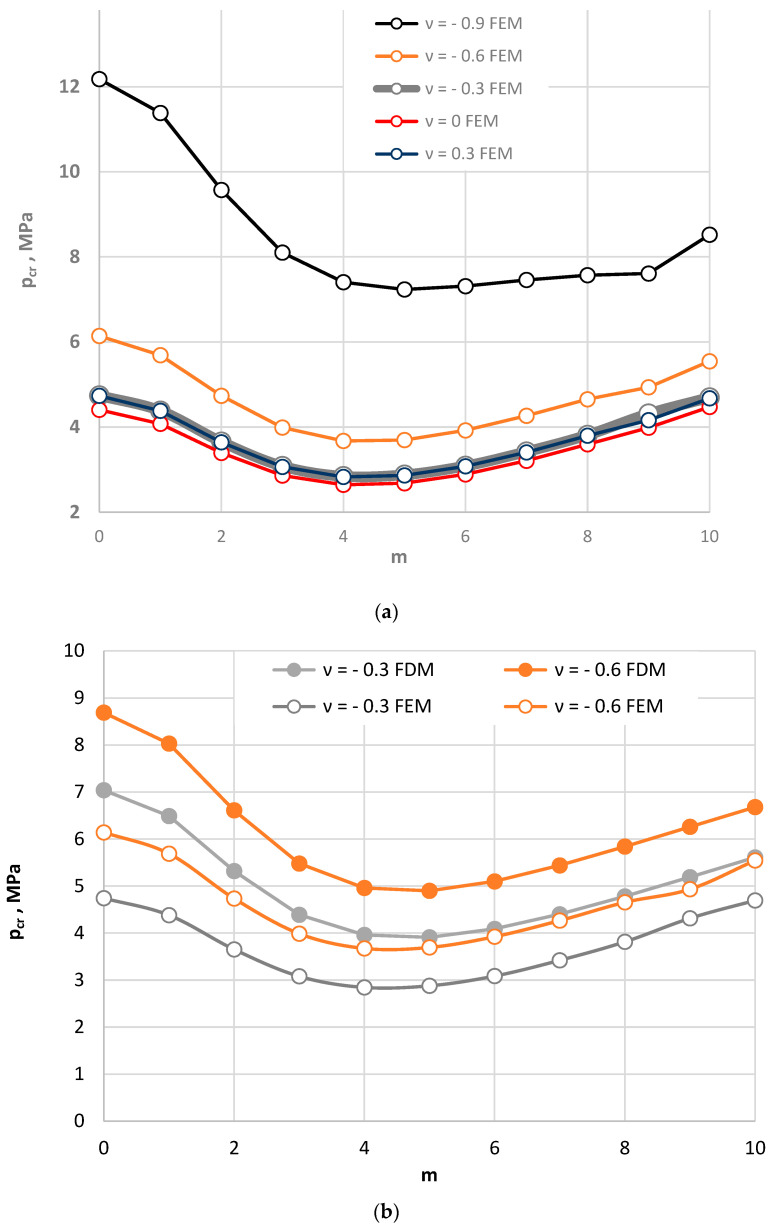
Critical static load *p_cr_* distribution versus mode *m* for three-layered plates radially compressed on the outer edge: FEM plate model (**a**), FDM and FEM plate model (**b**).

**Figure 10 materials-15-03579-f010:**
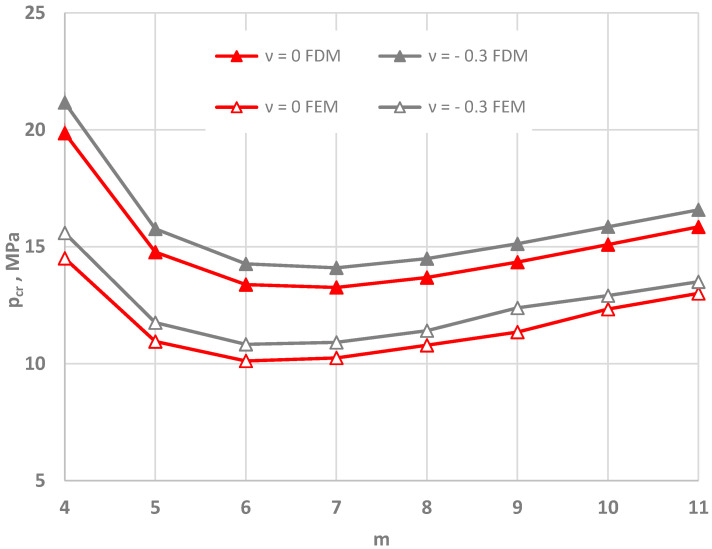
Critical static load *p_cr_* distribution versus mode *m* for the FDM and FEM three-layered plate models radially stretched on the inner edge.

**Figure 11 materials-15-03579-f011:**
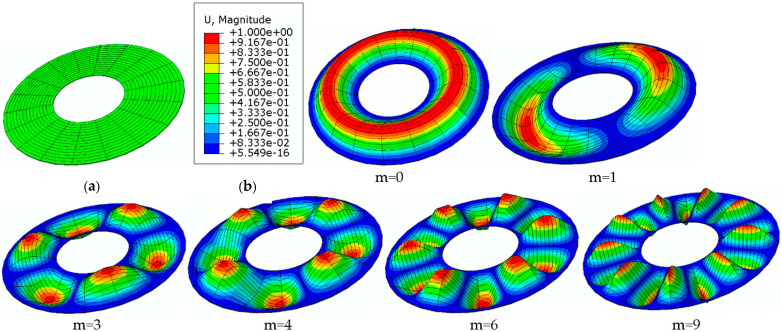
Three-layered annular FEM plate model: undeformed view (**a**), modes *m* of a plate radially compressed on the outer edge with auxetic facings *ν* = −0.6 (**b**).

**Table 1 materials-15-03579-t001:** Critical static load *p_cr_* of the composite plate model with Poisson’s ratio *ν* = −0.6 for auxetic facings radially compressed on the outer edge versus different numbers of discrete points *N*.

*m*	*p_cr_* [MPa]				
	*N* = 11	*N* = 14	*N* = 17	*N* = 21	*N* = 26
0	9.01	8.69	8.52	8.39	8.30
1	8.31	8.03	7.88	7.77	7.69
2	6.80	6.61	6.50	6.43	6.37
3	5.59	5.48	5.41	5.37	5.34
4	5.03	4.96	4.92	4.89	4.88
**5**	**4.95**	**4.90**	**4.87**	**4.86**	**4.85**
6	5.13	5.10	5.08	5.07	5.07
7	5.46	5.44	5.43	5.42	5.42
8	5.86	5.84	5.83	5.83	5.82
9	6.27	6.26	6.25	6.25	6.25
10	6.69	6.68	6.67	6.67	6.67

**Table 2 materials-15-03579-t002:** Critical static load *p_cr_* of the composite plate model with Poisson’s ratio *ν* = −0.9 for auxetic facings radially compressed on the inner edge versus a different numbers of discrete points *N*.

**number *N***	11	14	17	21	26
***p_cr_* [MPa]**	24.65	25.20	25.66	25.90	25.99

**Table 3 materials-15-03579-t003:** Critical static load *p_cr_* of the composite plate model with Poisson’s ratio *ν* = −0.3 for auxetic facings radially stretched on the inner edge versus a different numbers of discrete points *N*.

*m*	*p_cr_* [MPa]				
	*N* = 11	*N* = 14	*N* = 17	*N* = 21	*N* = 26
3	48.00	47.00	46.46	46.08	45.85
4	21.52	21.17	20.98	20.85	20.76
5	15.94	15.76	15.67	15.61	15.57
6	14.37	14.27	14.22	14.18	14.16
**7**	**14.16**	**14.10**	**14.07**	**14.05**	**14.04**
8	14.53	14.49	14.48	14.47	14.46
9	15.15	15.13	15.12	15.12	15.12
10	15.87	15.85	15.85	15.85	15.86

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
