# Peer review of "Static Stability of Composite Annular Plates with Auxetic Properties"

_materials, 2022, doi:10.3390/ma15103579_

Round 1
Reviewer 1 Report
The authors present a theoretical approach, using the finite difference method, to the evaluation of the static stability of complex, composite annular plates with layers with auxetic properties. The results are compared with plates with traditional facings, and with a finite element plate model. The plates were loaded radially, in compression and tension. The main impression is that this is an interesting study with merit. However, more work is needed before publication. One main issue with this manuscript is the language, which is somewhat awkward with unfamiliar wording (e.g. “slideably clamped”), and at times difficult to read. Furthermore, the state-of-the-art in the introduction is also written in an untraditional way. This could be re-written, and also include the authors of the references. Additionally, it is somewhat difficult to follow the results and discussions in Section 4. I think this could be helped by a more extensive problem formulation (Section 2). Here, you could explain/define the buckling issues that is discussed in Section 4 (m, N and so on). Please consider the following comments and questions:
- In this paper, I think it would be very beneficial with a list of notation in the beginning.
- Figure 1 is small and somewhat unclear/difficult to read. Also, a figure/sketch of the full plate would be helpful.
- Page 5 – what is meant by G, G2?
- Have you taken into account that G also can depend on Poisson’s ratio?
- It is not clear what m (is at buckling mode? Some definition/figure is needed – maybe in Section 2?) and N (which discrete points are we talking about) are.
Author Response
I would like to thank for all presented, very valuable comments presented in reviews.
According to them the manuscript has been changed in many places (they are written in red colour):
- one of the main observations has been added to the abstract,
- the introduction point has been expanded with the new works,
- Figure 1 has been improved,
- the list of notation has been added,
- the explanations have been added, for example connected with the name of the support system – “slideably” clamped – this determination has been used earlier in my works,
- according to the important note about the value of the Kirchhoff’s modulus the explanation has been presented – the value of it is not constant, calculations were carried out for the various values of Poisson’s ratio for the fixed value of Young’s modulus,
- the exemplary plate modes are shown in Fig.8 for homogeneous plate and Fig.10 for the three-layered plate,
- the calculations are complemented with the presentation of the some numbers relations between results.
The English language has been corrected by the professional office.

Reviewer 2 Report
### REVIEW REPORT ###
MANUSCRIPT ID: materials-1673886
MANUSCRIPT TITLE: Static stability of composite annular plate with auxetic properties
RECOMMENDATION: Major revision
SUMMARY: In the paper, the author introduces a numerical approach for investigation of static stability of composite annular plate with auxetic core. The author investigates an annular panel with two linear-elastic penetration resistant covers filled with an auxetic core. The author analyzes static stability of the problem and compares the results with the finite element method simulation. The paper represents and important and relevant topic in the field of materials with negative Poisson's ratio. However, I identified some problems regarding clarity, motivation and main message of the contribution. Thus, the paper is worth publishing after a major revision.
COMMENTS:
1) English language - professional language correction is strongly recommended. I went through problems with articles and the author sometimes uses hard-to-understand expressions or statements that are not grammatically correct.
2) Introduction and literature overview - clear and easy-to-understand motivation is missing in the introductory paragraph. Literature overview is oriented almost exclusively on a numerical case studies. References to experimental work or full-scale simulations would justify and clarify the scope and content of this study. For me, it was hard to understand what is the main step forward using the presented approach.
3) Problem formulation - is this really a problem formulation? It is necessary to formulate the problem based on the motivation. Why is particularly the composite, three-layered annular plate loaded mechanically on inner or outer edge the object of the analysis?
4) Problem formulation - boundary conditions - why is only the clamping boundary used?
5) Figure 1 - use different edge thicknesses or colors for better readability.
6) Problem solution - The solution is based on the eigenvalue problem, where the numerically calculated minimal values of loads correspond to the critical, static loads. Is this relevant for the referenced study, for the actual study or for both?
7) Elements of solution - the items represent the individual steps of the analysis? I recommend to re-write this part for better clarity.
9) Results discussion - this section is not a discussion of results as neither result has been presented so far.
10) Convergence evaluation - how was the simulation done? MATLAB based scripting? Custom code?
11) Results - what are the main benefits of the presented method? It is not discussed in the paper and represents its weakest point. Can be, e.g., complexity and time requirements for the presented method and for the FEM simulation compared?
12) Similar with the previous point. What are the main benefits of the presented method?
Author Response
I would like to thank for all presented, very valuable comments presented in reviews.
According to them the manuscript has been changed in many places (they are written in red colour):
- one of the main observations has been added to the abstract,
- the introduction point has been expanded with the new works,
- Figure 1 has been improved,
- the list of notation has been added,
- the explanations have been added, for example connected with the name of the support system – “slideably” clamped – this determination has been used earlier in my works,
- according to the important note about the value of the Kirchhoff’s modulus the explanation has been presented – the value of it is not constant, calculations were carried out for the various values of Poisson’s ratio for the fixed value of Young’s modulus,
- the exemplary plate modes are shown in Fig.8 for homogeneous plate and Fig.10 for the three-layered plate,
- the calculations are complemented with the presentation of the some numbers relations between results.
Additionally, I would like to explain that:
- that slideably clamped edges as the basic plate support system have been analysed. Of course, it can be extended on the others. Analytical and numerical solution has been performed using the author programme, which includes other kinds of support system,
- the evaluation of the minimal value of the static loads, which determine the plate critical state is the most important in stability problem. So, the eigenvalue problem, which was used in solution process enables such analysis.
The English language has been corrected by the professional office.

Reviewer 3 Report
“Paper presents the approach” please verify the English everywhere in this paper. Even at the very beginning noted some issue -as indicated
Overall the abstract is very verbally and vague – it don’t say nothing quantifiable – please revise it !
There is need for an appropriate introduction – now looks that the introduction just discusses 7 papers. I have no noted noting about the context of this work, what about challenges that you want to solve, the literature review should be presented in a critical manner, authors contribution and scientific novelty should be better highlighted !
You said “problem formulation “ but actually there is details of the investigated configuration !
What do you mean by problem solution- better saying strategy to solve the case study or something similar
The introductory part in results discussion should be moved in a section of strategy or method
You bring out a section about verification evaluation but nothing about the models designed and constraint used please develop to fully be able to replicate it
As suggested about abstract the same apply for conclusion they are too verbally – therefore I suggested rephrasing them using quantitative details from the results
The literature review list is too short and also all the references are out of date- please improve it with novel references
Author Response

(The authors gave the same response as above.)

Round 2
Reviewer 1 Report
The author has addressed many of the comments of the reviewers, and this is appreciated. However, it is not easy to see all changes and comments from the reply. Furthermore, on the more general level, which all of the reviewers comment on, the manuscript is still lacking. First of all, the author must make the motivation, problem formulation and discussion clearer. Second, the state-of-the-art art needs editing. It would be preferrable if the authors are mentioned, and the references are not called paper 1, 2 and so on. Again, a figure/sketch of the full plate in the problem formulation would be helpful although this is now shown later in the manuscript
Author Response
I would like to kindly thank for all presented remarks.
The motivation as a searching of a new possibilities of new composite structures has been presented in the introduction point of paper. The three research questions are formulated.
The form of reference presentation is changed. There are authors’ names.
The plate is presented by the general sketch and two a little improved cross-sections. I hope it is better.
The additional summary, which is connected with the formulated research questions, particularly with the evaluation of two plate models is presented in the conclusion point of work.

Reviewer 2 Report
### REVIEW REPORT ###
MANUSCRIPT ID: materials-1673886, revision 1
MANUSCRIPT TITLE: Static stability of composite annular plate with auxetic properties
RECOMMENDATION: Major revision
The cover letter summarizing the changes performed in the manuscript is vague. I really miss the point-by-point reactions on the individual comments of all the reviewers. In this current form, it is really hard to track the modifications although they are highlighted in red color. I was not able to identify changes according to the following points:
1) Motivation - clear and easy-to-understand motivation is still missing in the introductory paragraph. It is hard to understand what is the main step forward using this numerical approach.
2) Figure 1 - I don't see better readability. It was probably modified, but the figure is in my opinion still not sufficiently readable.
3) Results discussion - this section is not a discussion of results as neither result has been presented so far. I would recommend to use conventional naming, e.g., Results and Discussion
4) Results - what are the main benefits of the presented method? It is not discussed in the paper and represents the weakest point of the paper. Can be, e.g., complexity and time requirements for the presented method and for the FEM simulation compared? Was this comment somehow addressed? I am not able to find the related information in the text.
5) Similar with the previous point. What are the main benefits of the presented method? It has to be clearly stated in the conclusions.
Author Response
I would like to kindly thank for all presented remarks.
- The motivation as a searching of a new possibilities of new composite structures has been presented in the introduction point of paper. The three research questions are formulated.
- The plate is presented by the general sketch and two a little improved cross-sections. I hope it is better.
- The point named result discussion now is named very simply as Exemplary results. It includes the information that, of course, only selected plate examples are examined and results are analysed.
- and 5. In general the evaluation of the technique of the solution to the problem is not the aim of the undertaken analysis. Rather some stability phenomenon of plates of plate with new, different structure is the subject of the consideration. However, in order to evaluate the correctness of the accepted methods of solution two plate models are used. Of course, it creates the possibility to characterize both numerical models, which in computational issues may be interested.
But, the additional summary, which is connected with the formulated research questions, particularly with the evaluation of two plate models is presented in the conclusion point of work.

Reviewer 3 Report
.
Author Response
I would like to kindly thank for all presented remarks.

Round 3
Reviewer 1 Report
The author has addressed the comments of the reviewers, and the only comment now is that the language has to be edited.
Reviewer 2 Report
The paper was modified accordingly. I recommend to accept it in present form.